# Linking Serine/Glycine Metabolism to Radiotherapy Resistance

**DOI:** 10.3390/cancers13061191

**Published:** 2021-03-10

**Authors:** Anaís Sánchez-Castillo, Marc Vooijs, Kim R. Kampen

**Affiliations:** 1Department of Radiation Oncology (MAASTRO), GROW School for Oncology and Developmental Biology, Maastricht University Medical Center, P.O. 616, 6200 MD Maastricht, The Netherlands; a.sanchezcastillo@maastrichtuniversity.nl; 2Laboratory for Disease Mechanisms in Cancer, Department of Oncology, KU Leuven and Leuven Cancer Institute (LKI), Herestraat 49, 3000 Leuven, Belgium; marc.vooijs@maastrichtuniversity.nl

**Keywords:** serine and glycine metabolism, PHGDH, SHMT, PSAT1, PSPH, redox homeostasis, DNA repair, hypoxia, cancer, radiotherapy, resistance

## Abstract

**Simple Summary:**

Hyperactivation of the de novo serine/glycine biosynthesis across different cancer types and its critical contribution in tumor initiation, progression, and therapy resistance indicate the relevance of serine/glycine metabolism-targeted therapies as therapeutic intervention in cancer. In this review, we specifically focus on the contribution of the de novo serine/glycine biosynthesis pathway to radioresistance. We provide a future perspective on how de novo serine/glycine biosynthesis inhibition and serine-free diets may improve the outcome of radiotherapy. Future research in this field is needed to better understand serine/glycine metabolic reprogramming of cancer cells in response to radiation and the influence of this pathway in the tumor microenvironment, which may provide the rationale for the optimal combination therapies.

**Abstract:**

The activation of de novo serine/glycine biosynthesis in a subset of tumors has been described as a major contributor to tumor pathogenesis, poor outcome, and treatment resistance. Amplifications and mutations of de novo serine/glycine biosynthesis enzymes can trigger pathway activation; however, a large group of cancers displays serine/glycine pathway overexpression induced by oncogenic drivers and unknown regulatory mechanisms. A better understanding of the regulatory network of de novo serine/glycine biosynthesis activation in cancer might be essential to unveil opportunities to target tumor heterogeneity and therapy resistance. In the current review, we describe how the activation of de novo serine/glycine biosynthesis in cancer is linked to treatment resistance and its implications in the clinic. To our knowledge, only a few studies have identified this pathway as metabolic reprogramming of cancer cells in response to radiation therapy. We propose an important contribution of de novo serine/glycine biosynthesis pathway activation to radioresistance by being involved in cancer cell viability and proliferation, maintenance of cancer stem cells (CSCs), and redox homeostasis under hypoxia and nutrient-deprived conditions. Current approaches for inhibition of the de novo serine/glycine biosynthesis pathway provide new opportunities for therapeutic intervention, which in combination with radiotherapy might be a promising strategy for tumor control and ultimately eradication. Further research is needed to gain molecular and mechanistic insight into the activation of this pathway in response to radiation therapy and to design sophisticated stratification methods to select patients that might benefit from serine/glycine metabolism-targeted therapies in combination with radiotherapy.

## 1. Introduction

Alterations in cellular metabolism are a hallmark of tumorigenesis [1]. Cancer cells undergo metabolic reprogramming to support a higher demand of energy and metabolites for continuous cell growth and proliferation. It enables them to adapt and respond to changes in nutrient and oxygen availability, as well as other metabolic stress conditions. Enhanced glycolysis and glutaminolysis have been extensively described in cancer cells [2,3,4,5]. Nonetheless, the non-essential amino acid serine has recently attracted renewed attention due to mounting evidence as a key metabolite in the development, progression, and maintenance of cancer cells [6,7,8,9,10,11]. The sources of serine include its uptake from the extracellular microenvironment, as well as de novo serine/glycine biosynthesis from glucose in the cytosol and mitochondria. Interestingly, it has been shown that certain tumors have increased activation of de novo serine/glycine biosynthesis, while healthy cells mostly rely on extracellular uptake of serine and glycine [12,13,14]. However, some tumors also remain highly dependent on serine uptake to support cancer cell proliferation, in which serine and glycine-deprived diets have resulted in delayed cancer progression in animal models [15,16]. In recent years, there has been a growing interest in the activation of de novo serine/glycine biosynthesis in cancer and its contribution to tumor pathogenesis, poor outcome, and treatment resistance [17,18,19,20].

The intrinsic and acquired radioresistance of tumor cells remains a critical obstacle for the curative treatment of cancer patients, ultimately leading to tumor relapse. It is well established that eradication of clonogenic or cancer stem cells (CSCs) is required to achieve local tumor control [21,22]. Therefore, further elucidation of mechanisms involved in their response to irradiation and the dynamics of the irradiated tumor microenvironment is required to improve the efficacy of radiotherapy and concurrent chemo-radiation.

We suggest that the activation of de novo serine/glycine biosynthesis during hypoxia, a general characteristic of solid tumors and the CSCs microenvironmental niches, might contribute to radiotherapy resistance by being involved in the maintenance of cancer stem cells (CSCs) and redox homeostasis. Nevertheless, only a few studies have investigated metabolic reprogramming of serine/glycine biosynthesis of cancer cells in response to irradiation. This knowledge might be exploited to develop predictive biomarkers and agents that interfere with radiotherapy treatment response in tumors that rely on de novo serine/glycine biosynthesis. In the current review, we will describe the clinical relevance of de novo serine/glycine biosynthesis activation in cancer and its association with therapy resistance, with a particular focus on radioresistance.

## 2. Activation of De Novo Serine/Glycine Biosynthesis in Cancer

### 2.1. De Novo Serine/Glycine Biosynthesis Pathway

In de novo serine/glycine biosynthesis, the glycolytic or gluconeogenic intermediate, 3-phosphoglycerate (3PG), is used as the precursor for the synthesis of serine, starting with the NAD^+^-dependent oxidation of 3PG into 3-phosphohydroxypyruvate (3PHP) by phosphoglycerate dehydrogenase (PHGDH). 3PHP is then converted into phosphoserine through a transamination reaction catalyzed by phosphoserine aminotransferase 1 (PSAT1). This reaction uses glutamate as nitrogen donor and produces α-ketoglutarate (α-KG), which is an intermediate of the tricarboxylic acids (TCA) cycle. Hence, the serine synthesis pathway also contributes to the anapleurotic replenishment of the TCA cycle via the PSAT1 reaction. Finally, phosphoserine phosphatase (PSPH) hydrolyzes the phosphoserine phosphate group to produce serine (Figure 1). Serine is then required for several biosynthetic and signaling pathways, including the synthesis of phospholipids and other amino acids such as glycine and cysteine. The reversible interconversion of serine and glycine is catalyzed by either the cytosolic or mitochondrial serine hydroxymethyltransferase (SHMT), SHMT1 or SHMT2, respectively [12,23]. Importantly, this reaction donates carbon units to the one-carbon metabolism network, comprised of folate and methionine cycles and required for the synthesis of protein, lipids, nucleic acids, and methylation reactions (Figure 1) [24,25,26,27]. Cysteine synthesis through the transsulfuration pathway is also connected to the methionine cycle via homocysteine. Serine condenses with homocysteine to generate cystathionine, which is then cleaved to produce α-KG and cysteine. In addition, both cysteine and glycine are essential players in controlling redox balance through the synthesis of the antioxidant glutathione (GSH) [8]. GSH is a tripeptide thiol antioxidant composed of glycine, cysteine, and glutamic acid and is one of the main regulators of reactive oxygen species (ROS) in cancer owing to its role as ROS scavenger. ROS plays a dual function in cancer, being important for cellular signaling pathways that induce tumorigenesis, adaptation to hypoxia, and at the same time, ROS can trigger oxidative stress and cell death [28,29,30,31,32,33]. For this reason, cancer cells increase ROS production and their antioxidant capacity to counteract damaging oxidative effects and sustain cell viability and growth. Moreover, de novo serine/glycine biosynthesis and downstream one-carbon metabolism modulate the NAD+/NADH and NADP+/NADPH ratios, maintaining NADH and NADPH reductive equivalents in cancer cells [34,35,36]. Increased NAD/NADH and NADP+/NADPH ratios have been found five and ten times higher in cancer cells compared to normal cells, respectively [37]. NADH is required for mitochondrial oxidative phosphorylation and ATP production, and NADPH provides reducing equivalents to generate reduced forms of antioxidants molecules, such as the reduction of glutathione disulfide (GSSC) to GSH, which uses NADPH as cofactor, or the maintenance of the reduced form of thioredoxin (TRX) by TXR reductase, which uses NADPH as an electron donor. Additionally, NADPH is also required during mitochondrial oxidative phosphorylation and for the synthesis of fatty acids and steroids [38].

### 2.2. De Novo Serine/Glycine Biosynthesis Activation in Cancer

The number of cancer types identified as dependent on de novo serine/glycine biosynthesis is growing. Activation of the de novo serine/glycine biosynthesis pathway has been observed in cancer patients both as a consequence of gene amplification and alterations of upstream regulators that promote enhanced enzyme overexpression.

The de novo serine/glycine biosynthesis pathway gained interest after the discovery of PHGDH amplification in melanoma and breast cancer, which was correlated with a worse prognosis. PHGDH-amplified breast and melanoma cancer cells are dependent on de novo serine/glycine biosynthesis for their proliferation. In addition, ectopic expression of PHGDH in mammary epithelial cells enhances their propensity for neoplastic transformation, disrupting acinar morphogenesis. Suppression of PHGDH expression results in a decrease of α-KG levels, consequently decreasing the replenishment of glutamate into the TCA cycle. In vivo studies have also shown that increased PHGDH expression promotes tumor progression in mouse models of melanoma and breast cancer [14,39].

Analysis of large-scale public genomic data from The Cancer Genomic Atlas (TCGA) (cBioPortal) shows alterations of serine/glycine biosynthesis pathway enzymes due to gene amplification, mutations, and increased mRNA expression for these enzymes in a significant number of patients in different cancer types (Figure 2). Collectively, alterations have been found in ~35% of lung cancers, in ~37% of melanomas, and 25% of glioblastoma (GBM) patients, in the latter with recurrent amplifications of *PSPH* gene in 12% of cases (TCGA copy number portal, *p* = 5.25 × 10^−61^) [40,41]. In contrast, some cancers, such as lung adenocarcinoma and breast cancer, are characterized by serine and glycine pathway enzyme overexpression rather than genetic alterations in these enzymes. These data suggest a major contribution of this pathway in tumor pathogenesis and warrants further investigation into the mechanisms that induce serine/glycine biosynthesis hyperactivation in cancer.

### 2.3. Regulatory Mechanisms of De Novo Serine/Glycine Biosynthesis in Cancer

De novo serine/glycine biosynthesis pathway is regulated by a complex network that includes energy and nutrient-sensing mechanisms, transcriptional and epigenetic regulation of gene expression, posttranslational modification of enzymes and metabolites, along with the regulation of the activity of key enzymes of this metabolic pathway [11]. In the last decade, mutations in oncogenes and tumor suppressor genes have been identified to induce metabolic adaptation of cancer cells through activation of serine/glycine biosynthesis, enhancing the response to the metabolic/oxidative stress induced by hypoxia or nutrient-deprivation conditions. For instance, serine/glycine biosynthesis enzymes are direct transcriptional targets of the MYC oncogene family, and overexpressing/amplified-MYC tumors have been found to be dependent on this metabolic pathway for survival under glucose/glutamine-deprived conditions. In human hepatocellular carcinoma (HCC), cMYC-induced PSPH upregulation is a determining factor for the oncogenic activity of cMYC, and an important predictor of HCC patients’ survival [42]. Similarly, activating KRAS^G12D^ mutations have been described as increasing the expression of serine synthesis pathway enzymes. In vivo studies have shown that *Kras^G12D/+^*-driven mouse models of pancreatic and intestinal cancers are less responsive to depletion of serine and glycine, indicating that some cancer cells overcome serine dependence under serine-deprivation conditions due to activation of de novo serine/glycine biosynthesis [16]. In non-small cell lung cancer (NSCLC), KRAS mutations lead to increased glycolysis, subsequently feeding the serine/glycine biosynthesis side branch via NFR2 upregulation. This metabolic dependency enables the increased production of NADH and GSH to control redox homeostasis [43]. The activating transcription factor (ATF4) is another critical regulator of serine/glycine biosynthesis in different cancer types. ATF4 is translationally upregulated upon endoplasmic reticulum (ER) stress or amino acid depletion through PERK or GCN2-dependent eIF2α phosphorylation, respectively [44]. Under serine starvation, GCN2-ATF4 pathway activation and low activity of PKM2 regulate de novo serine/glycine biosynthesis and support cancer cell proliferation [45]. Serine allosterically regulates PKM2 to maintain serine homeostasis. Elevated levels of serine activate PKM2, which catalyzes the conversion of phosphoenolpyruvate into pyruvate in the last step of the glycolysis, restricting 3-PG channeling into de novo serine biosynthesis [46]. On the other hand, serine starvation results in low activity of PKM2 and subsequent accumulation of glycolytic intermediates. In addition, low serine levels lead to activation of GCN2 and enhanced translation of ATF4, which increases the transcription of de novo serine biosynthesis genes and the flux of glycolytic intermediates upstream of PKM2 into de novo serine/glycine biosynthesis [45]. In ER-negative breast cancer, ATF4 promotes the expression of *PSAT1* and enhances cell proliferation through the activation of GSK-3β/β-catenin/cyclin D1. Notably, in these cancers, *PSAT1* is upregulated and correlates with poor patient prognosis [47]. In NSCLC, mutations in *NRF2* have been demonstrated to induce the upregulation of PHGDH, PSAT1, and SHMT2 enzymes in an ATF4-dependent manner [17]. In neuroblastoma, MYCN and ATF4 act through a positive feedback loop. MYCN transcriptionally increases ATF4 expression, which in turn stabilizes MYCN protein levels by blocking FBXW7-mediated MYCN ubiquitination and further degradation, and, thereby, accelerating de novo serine/glycine biosynthesis [48].

Posttranslational modifications are also involved in the regulation of serine/glycine biosynthesis enzymes. A recent study has identified the interleukin enhancer-binding factor 3 (ILF3), which is overexpressed in primary colorectal cancer (CRC) patient samples and correlated with poor prognosis, to support the mRNA stability of serine/glycine biosynthesis genes. The described mechanism involves the EGF–EGFR–MEK–ERK pathway, in which ERK2 mediated-ILF3 S382 phosphorylation impedes the binding of the E3 ligase SPOP, thereby suppressing SPOP-mediated poly-ubiquitination and degradation of ILF3. In addition, it has been demonstrated that the metabolic reprogramming towards de novo serine/glycine biosynthesis due to ILF3 overexpression in CRC promotes tumor growth in vivo and the formation of patient-derived organoids (PDO) and patient-derived xenografts (PDX) [49].

Epigenetic mechanisms have also been identified to contribute to serine/glycine biosynthesis in cancer. The crosstalk between cellular metabolism and epigenetics is important in metabolic reprogramming and tumorigenesis. [50,51]. Epigenetic mechanisms that regulate serine/glycine biosynthesis enzymes include the histone H3 lysine 9 (H3K9) methyltransferase G9A, also known as EHMT2, the histone lysine demethylase KDM4C, and the E3 ubiquitin ligase MDM2. EHMT2 catalyzes the active chromatin mark H3K9 monomethylation, while KDM4C removes the repressive histone modification H3K9 trimethylation [52,53]. KDM4C controls the transcription of ATF4, which sequentially interacts with KDM4C to regulate the gene expression of serine/glycine biosynthesis pathway enzymes [53]. Finally, MDM2, a key negative regulator of the tumor suppressor TP53, is recruited to the chromatin and activates de novo serine/glycine biosynthesis in an ATF3 and ATF4-dependent manner, independently of p53. This mechanism is triggered by oxidative stress and low levels of the pyruvate kinase M2 (PKM2), enhancing flux into the serine/glycine biosynthesis pathway to replenish serine/glycine availability and generate reductive equivalents [54]. Liposarcomas are characterized by amplification of MDM2, which mediates serine metabolism addiction, promoting nucleotide synthesis to sustain tumor growth. Inhibition of MDM2 recruitment and binding to the chromatin by SP141 interferes with the MDM2-mediated regulation of de novo serine/glycine biosynthesis. Targeting the MDM2-mediated regulation of serine metabolism or endogenous PHGDH impairs cancer cell survival and proliferation, both in vitro and PDX models. In addition, serine/glycine-deprived conditions enhanced the antitumor effects of MDM2 and PHGDH inhibition. Hence, this data identifies MDM2 and serine metabolism dependency as potential therapeutic targets for liposarcoma [55].

Overall, increased serine/glycine metabolism might be beneficial for cancer cells by promoting purine and pyrimidine synthesis to support cancer cell viability; maintaining the nucleotide pool for sustained growth and proliferation; increasing antioxidant capacity and limiting ROS; as well as promoting DNA and histone methylation reactions.

## 3. Activation of De Novo Serine/Glycine Biosynthesis and Treatment Resistance

Reprogramming of cancer cell metabolism is associated with treatment resistance, including radio-, chemo-, and immunotherapy, as well as targeted therapies. For instance, increased glycolysis and glutaminolysis support breast cancer cells’ survival under genotoxic stress, promoting DNA repair by error-prone non-homologous end-joining (NHEJ) and preventing accelerated senescence after irradiation [56]. In vivo experiments with melanoma patient-derived cell lines and prospective analysis of patient samples showed that oxidative metabolism and hypoxia relate to decreased antitumor immunity and resistance to PD-1 blockade therapy [57].

### 3.1. Targeted Therapies

Activation of serine/glycine biosynthesis has been described as a mechanism of vemurafenib intrinsic and acquired resistance in melanoma, pancreatic, and NSCLC cancer cells [19]. Vemurafenib is an antagonist of the BRAF V600E mutation and disrupts downstream MAPK signaling pathway activation. Vemurafenib-resistant melanoma cells switch to oxidative metabolism during cell proliferation and become addicted to glutamine instead of glucose for proliferation [58]. Interestingly, siRNA silencing of PHGDH reverses acquired resistance to vemurafenib. In NRAS-mutant melanoma patients, phase I/II trials of MEK inhibitors have failed to improve the clinical outcome due to the development of resistance to the treatment, which may be linked to induced serine/glycine metabolism [59,60]. A recent study has shown upregulation of PHGDH in NRAS-mutant tumor melanoma PDXs with acquired resistance to MEK inhibitor PD901. Inhibition of PHGDH re-sensitizes resistant cells to MEK inhibitor, leading to oxidative stress and reduced cell growth [61]. In addition, PHGDH has been described as a key driver of the acquired resistance to the EGFR inhibitor erlotinib, which remains one of the major challenges in the targeted therapy of EGFR-activating mutation in lung adenocarcinoma (LUAD). In vitro and in vivo studies have shown that EGFR activation promotes serine synthesis [62]. The erlotinib-resistant cells showed an upregulation of PHGDH and resistance to oxidative stress with an increase in GSH production. Inhibition of PHGDH re-sensitizes resistant cells to erlotinib treatment. This data indicates that the inhibition of PHGDH might be a promising strategy to overcome the acquired erlotinib resistance in LUAD [63]. In hepatocellular carcinoma (HCC), high intracellular serine has been identified as a common metabolic feature of kinase inhibitor resistance to sorafenib, erlotinib, U0126, among other MEK inhibitors, being an important contributor of activation of the MAPK pathway via ERK phosphorylation upon glutamine withdrawal [64,65].

In multiple myeloma (MM), PHGDH and PSPH are suggested as potential predictive biomarkers to estimate the response to the proteasome inhibitor bortezomib (BTZ), which is the first-line treatment in MM [66,67]. However, an increase in the number of patients that acquire BTZ resistance has been reported [68,69]. Increased PHGDH and PSPH expression was observed in cells of MM patients who relapsed under BTZ treatment. However, larger patient numbers are required to further support the use of PHGDH and PSPH as predictive biomarkers for BTZ efficacy [66]. In addition, PHGDH has been described as a candidate to induce resistance to the HIF2α antagonists PT2385 and PT2399, which are being evaluated in phase I clinical trials for advanced or metastatic clear cell renal cell carcinoma. HIF2α deficient cells showed upregulation of de novo serine/glycine biosynthesis pathway, and inhibition of PHGDH reduced the growth of HIF2α-deficient tumor cells in vivo and in vitro [70].

### 3.2. Chemotherapy

De novo serine/glycine biosynthesis pathway might be a relevant target to improve the outcome of breast cancer patients that receive chemotherapy. PHGDH knockdown sensitizes both ER^+^ and ER^−^ breast cancer cell lines to carboplatin, increasing mitochondrial ROS, apoptosis, and abrogation of chemotherapy-induced breast CSCs enrichment [71]. Triple-negative breast cancer (TNBC) cells upregulate PHGDH upon exposure to doxorubicin as a mechanism to prevent the doxorubicin-induced formation of ROS. Repression of PHGDH, both in vitro and in vivo, increases the oxidative stress in TNBC cells and, therefore, improves the efficacy of doxorubicin treatment in TNBC [72].

The characterization of metabolic phenotypes of over 80 NSCLC cell lines identified a link between de novo serine/glycine biosynthesis and the prediction of pemetrexed sensitivity, which inhibits several folate-dependent reactions from one-carbon metabolism, i.e., thymidylate synthase and dihydrofolate reductase-catalyzed reactions [73] and binds and inhibit SHMT1 in vitro [74].

Serine biosynthesis inhibition and dietary restriction of serine have been shown to enhance the antitumor activity of the chemotherapeutic drug 5-fluorouracil, 5-FU. 5-FU is an analogue of the pyrimidine uracil, which functions by being incorporated into RNA and DNA, and inhibiting thymidylate synthetase, a key enzyme for deoxythymidine monophosphate (dTMP) production. dMTP is essential for DNA replication and repair, and, therefore, interference by 5-FU ultimately leads to cell death [75,76]. 5-FU resistance has been linked to PSAT1 overexpression since combined CRISPR-based deletion of PSAT1 and serine withdrawal suppressed CRC xenograft tumor growth and enhanced the sensitivity to 5-FU by increasing DNA damage and diminishing the nucleotide pool, resulting in cell death. Thus, this study suggests that both endogenous and exogenous serine contributes to CRC growth and resistance to 5-FU [77].

Altogether, these studies describe the contribution of de novo serine/glycine biosynthesis as a new mechanism of therapeutic resistance in response to chemotherapeutic drugs in breast, lung cancer, and CRC, as well as targeted therapies in tumors with abnormal activation of RAS–RAF–MEK–ERK pathway. In addition, the upregulation of serine/glycine biosynthesis enzymes has been proposed as predictive biomarkers for therapeutic responses in MM and clear cell renal cell carcinoma (Figure 3).

## 4. Radiotherapy Resistance and the Link with De Novo Serine/Glycine Biosynthesis

More than 60% of cancer patients receive radiotherapy as part of curative or palliative cancer management. Irradiation causes cell death as a result of oxidative stress due to the generation of ROS, unrepaired DNA damage, chromosomal instability, and the activation of cell stress response in the endoplasmic reticulum (ER) and the mitochondria. Nevertheless, radiotherapy is limited by the tolerance of normal tissues to long-term radiation-induced tissue damage, leading to de-escalation of the delivered dose and, therefore, hampering tumor control, together with the intrinsic and acquired radioresistance of tumor cells [78,79].

Radiation induces multiple cellular responses, including apoptosis, autophagy, senescence, cell cycle arrest, and DNA repair, as well as long-term irreversible tissue remodeling [80,81,82]. Besides, radiation affects metabolism [83]. Adaptive metabolic reprogramming of cancer cells, e.g., enhanced glycolysis or mitochondrial metabolism, can influence the outcome of radiotherapy treatment [84,85,86,87,88,89]. For instance, radioresistant cancer cells can undergo metabolic reprogramming towards glutamine anabolism via glutamine synthase (GS), which synthesizes glutamine from glutamate, and is transcriptionally regulated by STAT5 in response to radiation. Knockdown of GS causes a delay in DNA repair due to a diminished nucleotide metabolism, which leads to enhanced radiosensitivity in vitro and in vivo. Thus, GS promotes nucleotide synthesis for efficient DNA repair and the consequent growth of cancer cells under radiation stress [90].

To our knowledge, de novo serine/glycine biosynthesis activation has only been related to radiotherapy resistance in head neck squamous cell carcinoma (HNSCC). The metabolic profile of HNSCC cell lines revealed that the more radioresistant cells increase serine/glycine metabolism, the methionine cycle, nicotinic and nicotinamide, and purine metabolism. This data implicates the metabolic reprogramming of radioresistant cells to control the redox homeostasis, DNA repair, and DNA methylation reactions upon irradiation [20]. Other studies have also shown radiation-induced alterations of serine and glycine levels in different cancer cell lines. For instance, after irradiation, the metabolome of B16 melanoma cells showed an increase in alanine, glutamate, glycine, and choline levels. This indicates a radiation-mediated enrichment of glycine, arginine, and taurine metabolism, as well as glycolysis and gluconeogenesis [91]. In the breast cancer cell line HCC1937, irradiation led to increased glycine levels, whereas other amino acids, including isoleucine, leucine, tyrosine, and proline, were depleted [92]. Moreover, an increase in the circulating levels of serine, leucine, and isoleucine after radiotherapy in serum was observed in breast cancer patients who received local radiotherapy following tumor resection [93].

In summary, these studies link increased serine/glycine metabolism to radioresistance of cancer cells, appointing the activation of serine/glycine metabolism as an adaptive mechanism of radioresistant cells for their survival and recovery during or after radiation. The following part of the review will describe further associations between the de novo serine/glycine biosynthesis pathway and radiotherapy resistance, focusing on CSCs, the tumor microenvironment, and DNA damage response mechanisms.

### 4.1. CSC Fate and Functions

CSCs have been described as major drivers of tumor heterogeneity, recurrence, and resistance to cancer treatment. CSCs are tumor-initiating cells that retain molecular mechanisms of normal stem cells, such as their high self-renewal or the ability of multi-lineage differentiation, as well as the activation of highly conserved signaling pathways involved in development and tissue homeostasis. Increasing evidence has shown that CSCs are pivotal contributors of radioresistance in many tumor types, including GBM, HNSCC, breast cancer, or pancreatic cancer owing to intrinsic protective mechanisms, as well as the acquisition of radioresistance due to further genetic and epigenetic alterations of CSCs. CSCs radioresistance has been associated with stemness maintenance, induction of epithelial-mesenchymal transition, increased DNA repair capacity, lower ROS production, and, therefore, reduced induction of DNA damage, as well as the acquisition of quiescent phenotypes [94,95,96,97,98].

An important aspect of serine/glycine metabolism is its contribution to the control of stem cell functions. A recent study has highlighted the importance of PHGDH for maintaining CSCs stemness and self-renewal. Silencing of PHGDH leads to reduced expression of master transcriptional regulators of self-renewal and pluripotency, i.e., Oct4, Nanog, Sox2, and Bmi1, and diminishes the capacity of tumorsphere formation in embryonal carcinoma stem cells, breast CSCs, and patient-derived brain CSCs. PHGDH inhibition promotes multi-lineage differentiation of embryonal carcinoma stem cells via ubiquitination and proteasomal degradation of Oct4 and positively regulating the stability of the differentiation marker β3-tubulin. Moreover, this study provided a link between embryonal carcinoma stem cell differentiation following PHGDH inhibition and the induction of Beclin-1 dependent autophagy and senescence of embryonal carcinoma stem cells [99]. In addition, Oct4/Sox2/Nanog bind and regulate PSAT1, which is required to maintain intracellular levels of αKG and determine the fate of mouse embryonal stem cells. PSAT1 levels are essential for mouse embryonal stem cell self-renewal and pluripotency, whereas the knockdown of PSAT1 down-regulates α-KG, which lowers DNA 5′-hydroxymethylcytosine (5′-hmC) and increases H3K9me3 levels, ultimately rewiring the transcriptional program to induce differentiation [100].

Moreover, serine/glycine and one-carbon metabolism-mediated metabolic reprogramming of prostate cancer cells has been observed, leading to the induction of the neuroendocrine prostate cancer subtype that is associated with aberrant epigenetic events and acquired expression of CSC markers. Neuroendocrine prostate tumors are more aggressive and linked to therapeutic resistance and poor prognosis [101,102,103]. In these tumors, the loss of tumor suppressor protein kinase C (PKC)λ/ι induces serine metabolism via the mTORC1/ATF4/PHGDH axis, increasing the flux through one-carbon metabolism. Inactivation of PKCλ/ι showed increased methyl-cytosine labelling from [methyl-^13^C] methionine, a measure of methyl groups donated from S-adenosyl methionine (SAM). SAM is a co-substrate for SAM-dependent methyltransferases, in which dysregulation leads to aberrant DNA methylation and acquisition of the neuroendocrine prostate cancer phenotype. The upregulation of intracellular SAM levels upon (PKC)λ/ι loss could be rescued by knockdown of PHGDH [104].

Conversely, a recent study has shown that activation of de novo serine biosynthesis and α-KG induce differentiation of epidermal stem cells. Premalignant epidermal stem cells, which are cells of origin for squamous cell carcinoma, suppress de novo serine synthesis by favoring mitochondrial pyruvate consumption, which reduces the conversion of pyruvate to lactate in the cytosol, and, therefore, limits NAD+ regeneration. NAD+ is required for glycolysis and a cofactor for PHGDH. These cells become dependent on exogenous serine for growth and self-renewal, and serine/glycine-free diet impairs tumor initiation in mice, which can be reversed by inhibiting αKG-dependent dioxygenases. In addition, low levels of extracellular serine activate de novo serine biosynthesis in wild-type epidermal stem cells, which in turn stimulates αKG-dependent dioxygenases, removing the repressive histone modification H3K27me3 and activating their differentiation program [105]. In CRC, αKG has also been described to induce the transcription of differentiation-associated genes and subsequently downregulates Wnt target genes [106].

Hence, these studies show the relevance of de novo serine biosynthesis and exogenous serine levels in stem cell fate, tumor initiation, and the induction of more aggressive and heterogenous cancer phenotypes, but highlight the need to gain better insight into the roles of serine metabolism in both cancer and normal stem cells and their response to treatment. This knowledge might contribute to elucidate the best cancer-type dependent approach, i.e., inhibition of de novo serine biosynthesis, serine-free diets, or combination strategies, to enhance the therapeutic ratio of current treatments, targeting CSCs functions and CSCs contribution to treatment resistance and tumor recurrence.

### 4.2. Tumor Microenvironment

The presence of areas with low oxygen tension is frequently observed in solid cancers and is accompanied by acidic pH and nutrient-deprived conditions. These regions have been defined as a significant limiting factor in the outcome of cancer patients and a key driver of radiotherapy resistance and recurrence. This is explained by the dependency on the availability of oxygen to cause irreversible DNA damage in cancer cells by ionizing irradiation.

Hypoxic niches have been shown to support CSCs survival due to their low production of ROS and the activation of hypoxia-induced factor (HIF)-mediated stemness programs, as well as metastatic and quiescent phenotypes [107,108,109,110]. Moreover, vascularization and angiogenesis are induced in response to hypoxic signaling via increased secretion of pro-angiogenic factors and recruitment of bone-marrow-derived cells, which in turn cause increased proliferation and hypoxia and immune infiltration, the latest displaying both pro-and antitumor functions [111,112,113,114].

Therefore, strategies targeting hypoxia and the induction of different signaling pathways in response to hypoxia are critical to improving the sensitivity to radio-, chemo-, and immunotherapy.

#### 4.2.1. Hypoxia and ROS Homeostasis

In addition to the HIF–dependent switch from oxidative to glycolytic metabolism in cancer cells, HIF has been shown to promote de novo serine/glycine biosynthesis. Initially, it was demonstrated that HIF induces the expression of the mitochondrial enzyme SHMT2. Under hypoxic stress, the activation of HIF1 in MYC overexpressing cells induced the expression of SHMT2, resulting in the production of NADPH from NADP+ and maintenance of redox control. The knockdown of SHMT2 reduced NADPH/NADP+ ratio, increased ROS levels, and triggered hypoxia-induced cell death that could be abrogated by the addition of the antioxidant N-acetyl cysteine in vitro. In vivo, SHMT2 knockdown decreased tumor growth. In MYC-amplified neuroblastomas, the correlation between SHMT2 and HIF1α was associated with poor patient prognosis, confirming the essential role of HIF1-induced SHMT2 regulation in maintaining therapy-resistant hypoxic cell survival [115]. Later, it was demonstrated that HIF1 and HIF2 coordinately regulate the expression of other enzymes of this metabolic pathway in addition to SHMT2, i.e., PHGDH, PSAT1, and PSPH, as well as the mitochondrial one-carbon enzymes, MTHF dehydrogenase 2 (MTHFD2) and 10-Formyl-THF Synthetase (MTHFD1L). MTHFD2 catalyzes the reaction of MTHF and NADP^+^ to generate formyl-MTHF and NADPH, and MTHFD1L converts formyl-THF into THF and formate. This study was performed in a panel of breast cancer cell lines derived from ER^+^, HER2^+^, and TNBC, wherein PHGDH and SHMT2 showed hypoxia-induced expression across all the analyzed cell lines [71]. PHGDH silencing revealed that PHGDH is required to maintain NADPH levels, mitochondrial redox homeostasis, and for the survival of hypoxic breast cancer cells. Furthermore, an increased expression of PHGDH and SHMT2 was observed in breast CSCs. PHGDH knockdown caused a reduction in the number of breast CSCs in vitro and orthotopic tumors, suggesting the requirement of PHGDH for the induction of a breast CSC phenotype and promoting tumor initiation and breast cancer metastasis [71]. Recently, the upregulation of *PHGDH* and *SHMT2* expression under hypoxic conditions has also been observed in certain GBM cell lines. This study has shown that high levels of PHGDH protect GBM cells from hypoxia-induced cell death and starvation conditions, sustaining redox homeostasis and maintaining NADPH/NADP+ ratio [116]. Likewise, serine/glycine metabolism was previously associated with the survival of glioma tumor cells within the hypoxic areas. In human GBM, SHMT2 was found to be highly expressed in pseudopalisading cells, leading to metabolic rewiring via suppression of PKM2 activity and limiting pyruvate entry into the TCA cycle. Hence, metabolic reprogramming via upregulation of SHMT2 provides a survival advantage of cancer cells under hypoxic conditions [18].

Interestingly, regulators of serine/glycine biosynthesis activation have been associated with radiotherapy resistance by protecting cells against oxidative stress. The glutaminase-2 enzyme (GLS2) has been implicated in the radiation response of cervical carcinomas. GLS2 is overexpressed in cervical cancer tissues of radioresistant patients, and in vitro GLS2 knockdown reverses the radioresistant phenotype of different cancer cell lines by reducing GSH and NADH levels, and, therefore, increasing intracellular ROS levels [117]. GLS2 is a key enzyme of glutaminolysis and catalyzes the conversion of glutamine in glutamate, which can be subsequently used in the reaction catalyzed by PSAT1 in the synthesis of serine. TP73 regulates the activation of serine/glycine biosynthesis in cancer cells through transcriptional upregulation of glutaminase-2 enzyme (GLS-2). The production of glutamate and serine results in the accumulation of GSH, triggering a protective mechanism against oxidative stress and promoting the proliferation and survival of cancer cells [118].

Another example; ATF4 is involved in the induction of serine/glycine biosynthesis in different cancer types, e.g., ER-negative breast cancer, NSCLC, or neuroblastoma, and promotes cancer cell survival under stress conditions, e.g., hypoxic and nutrient-deprived environments [119]. Due to the requirement of oxygen for correct protein folding, severe hypoxia results in the activation of the unfolded protein response (UPR) that consists of three independent branches and leads to ATF6, IRE1, and PERK signaling. Downstream PERK-eIF2α-ATF4 signaling is essential for maintaining hypoxic cell survival and results in resistance of tumors to irradiation [120,121]. In GBM, radiation-induced ER stress signaling triggered PERK activity, which led to eIF2α phosphorylation, inducing an adaptive survival mechanism mediated by ATF4 upregulation [122].

#### 4.2.2. Immunomodulatory and Angiogenic Functions

Radiation elicits the activation of interconnected processes in the tumor microenvironment, including inflammation, cycling hypoxia, and modulation of cancer-associated fibroblasts, which subsequently lead to immunomodulation, re-vascularization, extracellular matrix remodeling, and fibrosis [123].

The immunomodulatory effects of radiation have been investigated to enhance the therapeutic ratio [124,125,126,127,128,129]. Radiotherapy displays both immune-stimulatory and suppressive functions, which can be influenced by the immune signature of the tumor microenvironment [130]. Radiation enhances the expansion and activation of T-cells in tumors with T-cell-rich microenvironment, while in immunosuppressive tumors, radiotherapy drives the recruitment of myeloid-derived suppressor cells (MDSC) populations and the polarization of macrophages from an inflammatory M1 phenotype into a pro-tumor M2 phenotype [131,132,133]. M2 macrophages have been associated with treatment resistance and relapse. For instance, M2 macrophages have been shown to contribute to vasculogenesis, leading to relapse of oral cancer following radiotherapy [134,135]. Strategies targeting macrophage infiltration and polarization in combination with radiotherapy have demonstrated synergy in preclinical models and early-phase clinical trials with enhanced antitumor efficacy [134,136,137,138]. Inverse data-driven modelling and multi-omics analysis have revealed that high PHGDH activity promotes M2 macrophage polarization [139]. Hence, the inhibition of de novo serine/glycine biosynthesis in cancer might reduce M2 macrophage polarization, contributing to the antitumor efficacy of these therapeutic strategies.

Furthermore, serine and glycine have been reported as immunosuppressive metabolites in inflammatory responses. Serine and glycine are required for the expression of the pro-inflammatory cytokine IL-1β induced by lipopolysaccharide (LPS) in macrophages. Macrophages increase mitochondrial ROS levels in response to LPS, and serine/glycine metabolism is required for the synthesis of GSH to maintain cellular redox homeostasis. Serine deprivation diminished LPS-induced IL-1β mRNA expression, and inhibition of PHGDH promoted survival of mice exposed to LPS. LPS has also been demonstrated to activate de novo serine/glycine biosynthesis and one-carbon metabolism for the synthesis of ATP and SAM during LPS-induced inflammation [140,141]. In the tumor microenvironment, macrophages produce IL-1β, which leads to the recruitment of myeloid cells from the bone marrow and their differentiation into immunosuppressive macrophages. Blocking IL-1β reversed the immunosuppressive tumor microenvironment of breast cancer mouse models and showed synergistic antitumor activity with anti-PD1 [142]. In addition, the upregulation of IL-1β upon irradiation and the subsequent activation of other inflammation-related molecules, e.g., matrix metalloproteinases that modulate the extracellular matrix, has been implicated in the development of fibrosis [143,144]. Therefore, serine deprivation may be an effective strategy to limit IL-1ß production, enhancing tumor control and reducing the cytotoxic effects on normal tissue.

The de novo serine/glycine biosynthesis has been shown to be essential for endothelial cell function, a relevant component of the tumor microenvironment involved in tumor growth and the response to radiotherapy [145]. PHGDH silencing impairs heme synthesis, which causes oxidative stress due to decreased GSH and NADPH synthesis and mitochondrial dysfunction. Heme groups are required for anchorage (complex II) or activity (complexes III and IV) of the electron transport chain (ETC) complexes, and insufficient heme production can compromise the activity of the enzymes of ETC, which ultimately leads to lethal angiogenesis defects [146,147]. Inhibition of serine/glycine biosynthesis may impair the formation of new blood vessels in the tumor microenvironment that contribute to tumor hypoxia, and, therefore, increasing the radiotherapy response. However, this function of serine/glycine biosynthesis in endothelial cells should be carefully considered in therapeutic strategies since its inhibition could lead to normal tissue toxicity.

### 4.3. DNA Damage Response (DDR)

Radiotherapy induces direct single and double-strand DNA breaks (SSB and DSB, respectively), but most of the DNA damage is caused by indirect DNA damage through the generation of ROS from H_2_O that modifies DNA. Many DNA modifications of ROS are reversible in the presence of oxygen but exacerbated under hypoxic conditions [148]. Irradiation-induced DNA lesions include oxidative DNA base modifications, crosslinking, as well as both SSB and DSB. The DDR mechanisms enable cells to detect DNA lesions, activate signaling pathways, and promote DNA repair. Therefore, pathways involved in signaling and repair of DNA strand breaks are critical for the outcome of radiation response.

Interestingly, the de novo serine/glycine biosynthesis pathway has been found to be activated under conditions of DNA damage and genomic instability. PHGDH, PSAT1, and PSPH protein levels increase in cells with persistent SSBs and in response to stress-induced conditions, together with one-carbon metabolism enzymes and proteins involved in amino acid uptake. These cells switch to an anabolic cellular state by enhancing their amino acid synthesis and increase their antioxidant capacity via GSH production [149]. Similarly, serine/glycine metabolism has been directly associated with DNA stability and maintenance in lung adenocarcinoma with poor prognosis by sensing and regulating the availability of purines and pyrimidines, which are essential components for DNA repair. Serine metabolism might function as a feeder pathway to control DNA synthesis. LUAD cellular models with high levels of PHGDH exhibit a proliferative and migratory phenotype due to the synthesis of pyrimidines and the production of GSH. In contrast, depletion of PHGDH limits the generation of sufficient amounts of these nucleotides leading to an accumulation of DNA strand breaks [150].

Cell cycle arrest facilitates DNA repair and the maintenance of genomic stability during DDR [151]. The tumor suppressor TP53 is one of the major effectors of the cellular response to radiation. The activation of p53-mediated signaling can induce cell cycle arrest, promoting cell survival, or apoptosis and senescence, leading to cell death. Therefore, p53 is an important determinant of radiation sensitivity [152]. In response to irradiation, p53 binds and activates the transcription of p21, which is an inhibitor of cyclin-dependent protein kinases that regulate cell entry into S-phase. It has been shown that serine starvation stimulates the recruitment of p53 to the p21 promoter and transiently activates the p53-p21 axis with the consequent induction of p21-dependent G1 cell cycle arrest. This is part of a survival response that allows the channeling of the limited levels of serine to GSH production instead of nucleotides synthesis, and, therefore, counteracting oxidative stress by preserving the cellular antioxidant capacity [153]. In addition, glycine deprivation and inhibition of SHMT2 expression have been demonstrated to reduce the proliferation of HeLa cells due to a prolonged G1 phase of the cell cycle [154]. Therefore, the link between serine/glycine metabolism and p53-p21 induction of cell cycle arrest to support cell survival and proliferation suggests the need to explore this axis in response to radiotherapy.

Collectively, we highlight a significant amount of evidence that de novo serine/glycine biosynthesis pathway activation might be involved in the responses to radiation therapy in cancer patients, which requires further investigation (Figure 4).

## 5. Future Perspectives: Therapeutic Targeting of De Novo Serine/Glycine Biosynthesis to Improve Radiotherapy Response

The discovery of the crucial role of serine/glycine metabolism in cancer progression and treatment resistance may provide new therapeutic intervention opportunities in combination with current cancer treatments. Radiotherapy is one of the main therapeutic options of cancer patients as part of their curative or palliative cancer management; however, little is known about the implication of the de novo serine/glycine biosynthesis pathway in the radiation therapy response. The upregulation of de novo serine/glycine biosynthesis under hypoxic and nutrient-deprived conditions and its relevance in cancer cell viability and proliferation, CSC maintenance, and the synthesis of reductive equivalents, which protect cancer cells against redox stress, suggest that the inhibition of de novo serine/glycine biosynthesis pathway in combination with radiotherapy might improve tumor control. Moreover, it is important to consider the roles of serine/glycine metabolism in the tumor microenvironment, e.g., in endothelial cells and the immune ecosystem in tumors.

The therapeutic strategies targeting serine/glycine metabolism include limiting the availability of exogenous serine and glycine in serine uptake-dependent cancers by diet restriction and targeting the enzymes of de novo serine/glycine biosynthesis pathway [16,77,155]. Therefore, it is essential to determine which cancer patients would benefit from these strategies. For instance, serine starvation resulted in reduced viability and impaired proliferation in p53-deficient tumors [153]. Inhibition of de novo serine/glycine biosynthesis is likely most effective in tumors that are reliant on de novo serine/glycine biosynthesis, such as PHGDH-amplified melanoma and breast cancers, as well as tumors with increased dependency on this pathway under metabolic/stress conditions, e.g., KRAS-mutated pancreatic tumors, which become insensitive to serine/glycine deprivation [16,26], or GBM and cMYC-amplified neuroblastomas with upregulation of this pathway in response to hypoxia [115,116]. However, it is important to consider the metabolic plasticity and the compensatory mechanisms following these therapeutic strategies. For instance, in cMYC-induced liver tumors, the inhibition of de novo serine synthesis via PSAT1 knockout was compensated by uptake of exogenous serine, in which serine and glycine deficient diet synergistically suppressed tumorigenesis [156]. Moreover, the therapeutic efficacy of combining serine/glycine withdrawal from the diet with PHGDH inhibition has been recently shown in preclinical models for liposarcoma and CRC [55,155]. In this example, reduced one-carbon metabolism cooperatively inhibits the growth of CRC tumors that are resistant to diet restriction or enzyme inhibition as individual therapies [155]. The sensitivity to de novo serine/glycine biosynthesis inhibitors may also be influenced by physiological levels of serine in the tumor tissue of origin. Some tumors arising in serine-limited environments adapt by upregulating de novo serine biosynthesis pathway enzymes and become more resistant to serine/glycine-targeted therapeutic interventions [157]. The availability of serine depends on the plasma metabolite levels and the capacity of tumors to obtain these nutrients, which is influenced, for instance, by tumor angiogenesis and metabolic competition with stromal and immune cells [158,159]. A limitation regarding dietary intervention’s applicability into the clinic includes the feasibility of total depletion of serine and glycine from the diet, which further support the need to consider the combination of dietary intervention and de novo serine/glycine biosynthesis inhibition to limit serine availability and improve the therapeutic outcomes of cancer patients.

Both serine-depleted diets and inhibitors of de novo serine/glycine biosynthesis have been combined with mitochondrial inhibitors, showing additive and synergistic antitumor effects. For instance, serine starvation has been combined with the biguanide antidiabetic drugs metformin and phenformin, which target ETC Complex I, consequently inhibiting mitochondrial oxidative phosphorylation [160,161]. Our group’s most recent discovery highlights the repurposed clinical anti-depressant sertraline as a dual SHMT1/2 inhibitor. Sertraline binds both SHMT1 and SHMT2 enzymes, acting as a competitive inhibitor for the folate binding pocket, and selectively targets different serine/glycine biosynthesis-dependent cancer models. Sertraline showed enhanced antitumor activity combined with compounds causing mitochondrial dysfunction, e.g., rotenone, antimycin A and the clinically used antimalarial artemether, in serine/glycine-dependent breast cancer mouse models [162]. The synergistic antitumor effect of these combinations is supported by the reciprocal relation between serine and one-carbon metabolism and mitochondrial functions. Overflow of serine-derived formate could contribute to oxidative phosphorylation in the mitochondria by producing NADH, a reducing equivalent used in the ETC [36]. In addition, serine availability plays a major role in maintaining mitochondrial metabolism by regulating ceramide and sphingolipid metabolism. Serine deficiency affects mitochondrial morphology and membrane potential and causes mitochondrial fragmentation, which is independent of nucleotide and redox metabolism [25]. On the other hand, respiratory chain dysfunction upon mitochondrial DNA depletion or lesions in the ETC impairs one-carbon metabolism [163]. Interestingly, targeting mitochondria function and metabolic reprogramming of cancer cells has also been suggested to increase radiotherapy responses [164]. For instance, several studies suggest that the use of metformin improves the response to radiation therapy in CRC, oesophageal cancer, NSCLC, and luminal breast cancer [165,166,167,168,169]. Using metformin treatment before irradiation increased tumor oxygenation in two CRC xenograft models. In addition, the use of metformin was associated with a better outcome for prostate cancer patients [165]. Metformin radio-sensitizes oesophageal cancer cells in vitro, increasing apoptosis, G0/G1 arrest, and AMPK activation [166,167]. In patients with oesophageal cancer, the use of metformin was associated with a dose-dependent increased response to concurrent chemo-radiation [170]. Metformin also inhibits the proliferation and tumor growth of NSCLC models and sensitizes NSCLC cells and tumors to irradiation, enhancing the radiation-mediated pro-apoptotic effects via ATM–AMPK pathway [168]. This therapeutic strategy, in combination with serine/glycine metabolism-targeted therapies, requires further detailed investigation on applicability to establish future clinical trials (Figure 5).

The rational integration of these combined treatment modalities is important in order to optimize the efficacy and enhance the therapeutic ratio of radiotherapy. Targeting serine/glycine metabolism before radiotherapy, concomitantly or sequentially, requires considering serine/glycine metabolism functions in cancer cells and the tumor microenvironment and how targeting this metabolic pathway might enhance or counteract the response to radiotherapy. For instance, before radiotherapy, a drug might be beneficial by reducing tumor hypoxia and increasing tumor oxygenation, thereby resulting in enhanced tumor kill. De novo serine/glycine biosynthesis inhibition before radiation might indirectly reduce tumor hypoxia by decreasing tumor cell proliferation and survival, contributing to the radiation effect. However, the loss of proliferation would, on the other hand, make cells less sensitive to radiation. Targeting de novo serine/glycine biosynthesis in combination with radiotherapy may enhance ROS generation via reduction of the antioxidant GSH and NAD(P)H, increasing DNA damage and limiting the nucleotides for DNA repair and proliferation. Blocking the induction of de novo serine/glycine biosynthesis under hypoxic conditions could radio-sensitize hypoxic cells and improve the eradication of CSCs by reducing their antioxidant mechanisms, as well as CSCs stemness and self-renewal. Finally, targeting de novo serine/glycine biosynthesis after radiotherapy might affect the viability and proliferation of remaining cancer cells, increasing the intracellular ROS levels and limiting the nucleotide pool to repair the DNA lesions, ultimately leading to apoptosis (Figure 5).

Regarding the tumor microenvironment, silencing de novo serine/glycine biosynthesis before radiotherapy may lead to angiogenic defects in endothelial cells, which might be beneficial to reduce the formation of new blood vessels, and, therefore, tumor hypoxia. However, it is important to consider the normal tissue toxicity that might be associated with the inhibition of this pathway in endothelial cells. In combination with radiotherapy, serine deprivation might diminish the radiation-induced upregulation of IL-1β in macrophages, which have been implicated in the recruitment of myeloid cells and their differentiation into immunosuppressive macrophages, as well as in the development of fibrosis. In addition, targeting de novo serine/glycine biosynthesis might decrease the switch to pro-tumor M2 macrophages. There are some limitations to consider, such as the requirement of serine/glycine uptake and SHMT for optimal T cell activation and proliferation via increased mitochondrial one-carbon metabolism [171,172]. Hence, targeting serine/glycine metabolism might affect the antitumor immune response in combination with radiotherapy (Figure 5). Yet, the requirement of serine and glycine for the antitumor immune response has not been studied so far.

A better understanding of serine/glycine metabolic reprogramming of cancer cells in response to radiation and the influence of this pathway in the tumor microenvironment may provide the rationale for the optimal combination of radiotherapy and serine/glycine metabolism-targeted therapies. The described studies in this review provide a starting point for discussion and future research in this field.

## Figures and Tables

**Figure 1 cancers-13-01191-f001:**
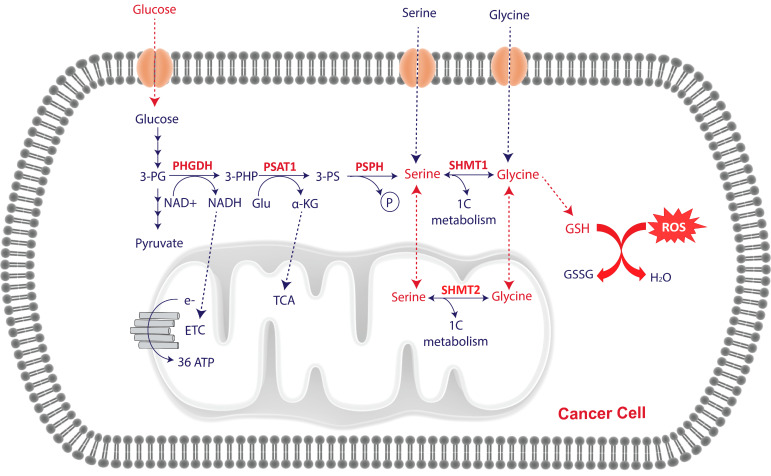
Overview of serine/glycine biosynthesis pathway activation in cancer. Serine/glycine biosynthesis pathway branches from the glycolysis, in which the intermediate 3P-glycerate is converted into serine and glycine after a chain of consecutive enzymatic reactions controlled by PHGDH, PSAT, PSPH, and SHMT1/2 enzymes. Cancer cells are characterized by activation of de novo serine/glycine biosynthesis to meet their high biosynthetic and energetic requirements. Activation of de novo serine/glycine biosynthesis and one-carbon (1C) metabolism promote purine and pyrimidine synthesis and supports lipid metabolism, as well as produces reductive equivalents to control redox homeostasis and α-ketoglutarate and S-adenosyl methionine (SAM) to regulate DNA and histone methylation. This representation shows the reduction of glutathione disulfide (GSSC) to glutathione (GSH), which is a tripeptide thiol antioxidant, composed of glycine, cysteine, and glutamic acid, and one of the main regulators of reactive oxygen species (ROS).

**Figure 2 cancers-13-01191-f002:**
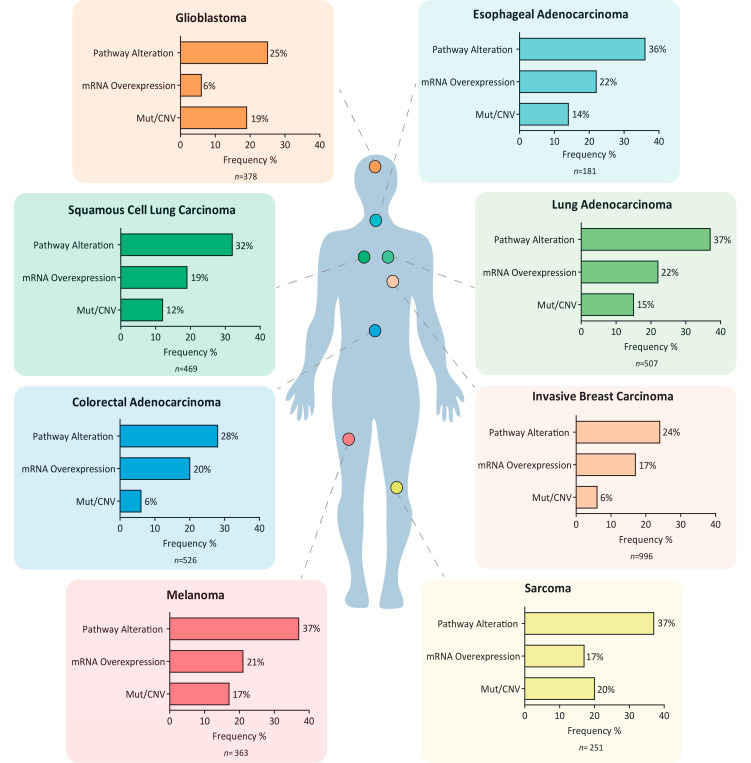
Serine/glycine biosynthesis pathway hyperactivation signatures in cancer. Representation of the frequency of distinct alterations in serine/glycine biosynthesis genes, i.e., *PHGDH*, *PSAT1*, *PSPH*, *SHMT1*, and *SHMT2*, across a set of patients for different cancer types. These alterations include mRNA overexpression, somatic mutations, and copy number gain/amplification, which can be associated with serine/glycine biosynthesis pathway hyperactivation signature. The graphs also show the frequencies of mRNA overexpression and mutations/copy number variations separately in each case. This data was collected from the The Cancer Genomic Atlas (TCGA) PanCancer Atlas datasets in cBioPortal.org (data collection July 2020).

**Figure 3 cancers-13-01191-f003:**
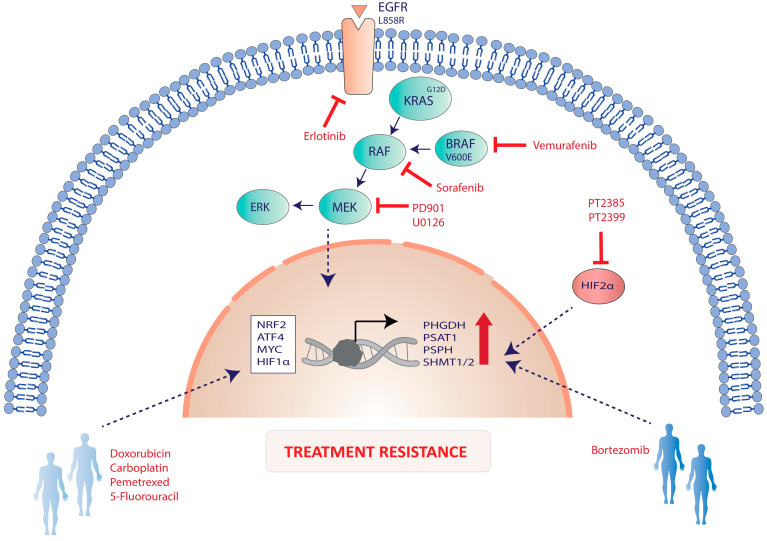
Serine/glycine biosynthesis pathway activation in tumor treatment resistance. The response to targeted therapies in tumors with abnormal activation of RAS–RAF–MEK–ERK pathway has been associated with increased expression of serine/glycine biosynthesis genes, which might be induced by oncogenic drivers such as EGFR^L858R^ and KRAS^G12D^ via ATF4 and NFR2, respectively. In addition, serine/glycine biosynthesis pathway hyperactivation has been linked to the resistance to chemotherapeutic drugs, e.g., doxorubicin, carboplatin, pemetrexed, or 5-fluorouracil in breast, lung, and colorectal cancer (CRC), the proteasome inhibitor bortezomib in multiple myeloma, as well as HIF2α antagonists in renal cell carcinoma.

**Figure 4 cancers-13-01191-f004:**
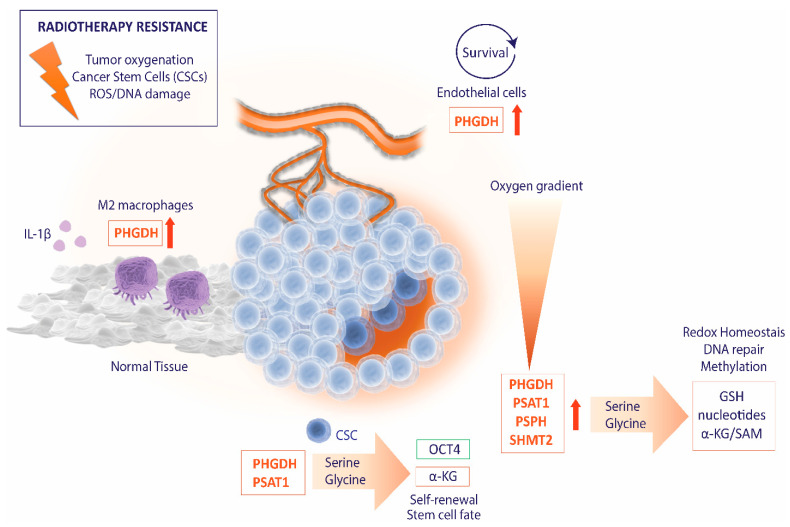
Linking radiotherapy resistance and serine/glycine metabolism in the tumor and associated tumor microenvironment. Activation of de novo serine/glycine biosynthesis pathway in cancer stem cells (CSC) under hypoxia and nutrient-deprived conditions is important to sustain cancer cell viability and proliferation, supporting the synthesis of nucleotides for DNA repair, as well as the synthesis of reductive equivalents that protect the cancer stem cells and their progeny against redox stress via autocrine and paracrine routes. In addition, de novo serine/glycine biosynthesis enzymes, i.e., PHGDH and PSAT1, are involved in the maintenance of CSCs stemness and self-renewal, for instance, via posttranslational regulation of OCT4, as well as regulating stem cell fate via αKG. In the tumor microenvironment, high PHGDH activity is essential for the survival of endothelial cells and promotes an immunosuppressive microenvironment with an increase in M2 macrophage polarization. In addition, serine and glycine are required for the expression of the pro-inflammatory cytokine IL-1β.

**Figure 5 cancers-13-01191-f005:**
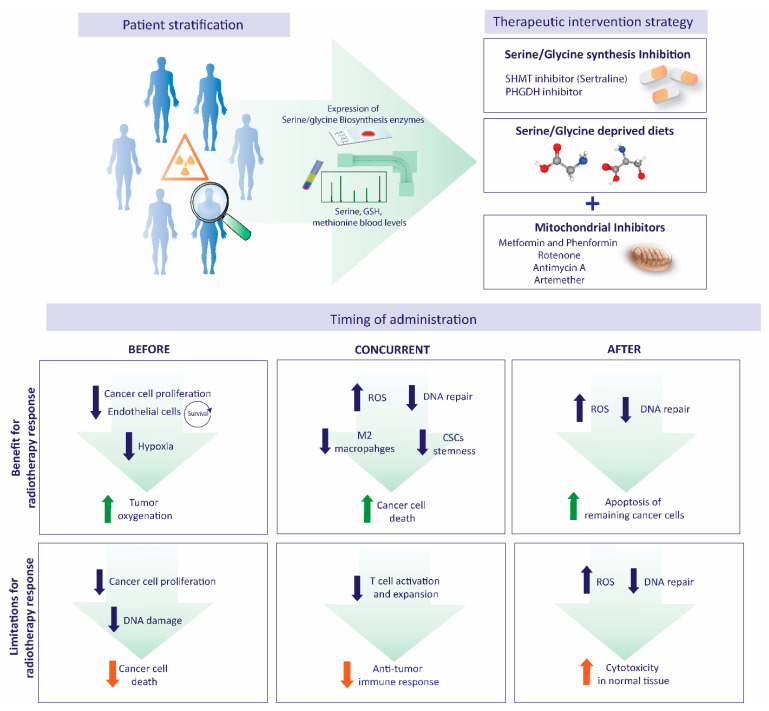
Future perspectives overview implicating serine/glycine metabolism-targeted therapies in combination with radiation therapy. The rational integration of serine/glycine metabolism-targeted therapies in combination with radiotherapy is important to optimize the efficacy and enhance the therapeutic ratio of radiotherapy. The stratification methods to select patients that might benefit from these combined treatment modalities could include the analysis of serine, GSH, or methionine serum levels or the expression of de novo serine/glycine biosynthesis enzymes. The therapeutic strategies targeting serine/glycine metabolism include limiting the availability of exogenous serine by diet restriction and targeting the enzymes of de novo serine/glycine biosynthesis pathways, for instance, SHMT inhibition using sertraline. In addition, mitochondrial inhibitors, e.g., metformin, phenformin or artemether, might have synergistic antitumor effects in combination with serine/glycine-targeted strategies and enhance the response to radiotherapy. Finally, targeting serine/glycine metabolism before radiotherapy, concomitantly or sequentially, requires considering serine/glycine metabolism functions in cancer cells and the tumor microenvironment and how targeting this metabolic pathway might enhance or counteract the response to radiotherapy. The blue arrows indicate expected cellular changes, in green beneficial outcome of the cellular changes and in orange the limitations of the outcomes related to the cellular changes.

## Data Availability

PubMed database was searched for research articles and reviews published before February 2021, using the following terms: Serine, Glycine, metabolism, PHGDH, SHMT, PSAT1, PSPH, redox homeostasis, DNA repair, hypoxia, cancer, radiotherapy, resistance. Data of mutations, copy number variations, and mRNA overexpression of de novo serine/glycine biosynthesis genes has been collected from cBioPortal (http://www.cbioportal.org/, accessed on 4 February 2021) data repositories using the publicly available TCGA (The Cancer Genome Atlas) PanCancer Atlas datasets.

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
