# Peer review of "Linking Serine/Glycine Metabolism to Radiotherapy Resistance"

_cancers, 2021, doi:10.3390/cancers13061191_

Round 1

Reviewer 1 Report

The authors reviewed the current understanding and the potential role(s) of serine and glycine metabolism in radiation resistance. This is an interesting and unique angle to develop new therapeutic strategy against aggressive cancer.  Here are a few suggestions:

  1. The key functions of serine metabolism pathway are not only making glycine, but producing one carbon unit, with concurrent generation of reducing equivalent and ATP. Glycine is never a limiting factor for tumor growth. In fact, high glycine levels inhibit tumor cell proliferation.  It is recommended to change the title to: ‘Linking serine and one carbon unit metabolism to Radiotherapy Resistance.’
  2. The authors cited multiple publications discussing the role of ATF4 in regulating serine biosynthesis enzymes but missed one earlier paper that showed ATF4 upregulates PHGDH, PSAT1 and PSPH1.
  3. Page 9, first paragraph. Although the αKG generated from serine synthesis may promote stemness, it is important to point out that there are multiple publications showing that αKG can induce differentiation in tumor cells 2-3.
  4. Page 9, second paragraph. There is no evidence showing that one carbon unit from serine is used for SAM synthesis. Ref 85 showed that (PKC)λ/ι KO increased M+1 mCyt labeling from 13C-methinonine, not from serine.  

References:

1. Ye, J.; Mancuso, A.; Tong, X.; Ward, P. S.; Fan, J.; Rabinowitz, J. D.; Thompson, C. B., Pyruvate kinase M2 promotes de novo serine synthesis to sustain mTORC1 activity and cell proliferation. Proc Natl Acad Sci U S A 2012, 109 (18), 6904-9.

2. Baksh, S. C.; Todorova, P. K.; Gur-Cohen, S.; Hurwitz, B.; Ge, Y.; Novak, J. S. S.; Tierney, M. T.; Dela Cruz-Racelis, J.; Fuchs, E.; Finley, L. W. S., Extracellular serine controls epidermal stem cell fate and tumour initiation. Nat Cell Biol 2020, 22 (7), 779-790.

3. Tran, T. Q.; Hanse, E. A.; Habowski, A. N.; Li, H.; Gabra, M. B. I.; Yang, Y.; Lowman, X. H.; Ooi, A. M.; Liao, S. Y.; Edwards, R. A.; Waterman, M. L.; Kong, M., alpha-Ketoglutarate attenuates Wnt signaling and drives differentiation in colorectal cancer. Nat Cancer 2020, 1 (3), 345-358.

Author Response

Reviewer 1

  1. The key functions of serine metabolism pathway are not only making glycine, but producing one carbon unit, with concurrent generation of reducing equivalent and ATP. Glycine is never a limiting factor for tumor growth. In fact, high glycine levels inhibit tumor cell proliferation. It is recommended to change the title to: ‘Linking serine and one carbon unit metabolism to Radiotherapy Resistance.’

We thank the reviewer for this suggestion. We have modified the title of the manuscript slightly into “ Linking Serine/Glycine metabolism to radiotherapy resistance.” We agree with the reviewer that serine supports one carbon metabolism, however, in this paper we also focus on downstream metabolites beyond one carbon metabolism, such as the effects of alpha-ketogluterate on methylation. Therefore, we find the suggested title an underestimation of all the interconnected metabolites discussed within this review. Additionally, we included the reference (15) that supports the statement mentioned by the reviewer:

  1. Labuschagne, C.F.; Van Den Broek, N.J.; Mackay, G.M.; Vousden, K.H.; Maddocks, O.D. Serine, but not glycine, supports one-carbon metabolism and proliferation of cancer cells. Cell reports 2014, 7, 1248-1258.
  2. The authors cited multiple publications discussing the role of ATF4 in regulating serine biosynthesis enzymes but missed one earlier paper that showed ATF4 upregulates PHGDH, PSAT1 and PSPH1.

We have included the paper, reference 45 in page 6, text line 202, as well as an additional paper (Chaneton, B. et al., 2012), reference 46 in page 6, text line 205, describing the feedback regulation of serine and pyruvate kinase M2, which is also related to the paper (Ye, J. et al, 2012) mentioned by the reviewer.

  1. Ye, J.; Mancuso, A.; Tong, X.; Ward, P.S.; Fan, J.; Rabinowitz, J.D.; Thompson, C.B. Pyruvate kinase M2 promotes de novo serine synthesis to sustain mTORC1 activity and cell proliferation. Proceedings of the National Academy of Sciences 2012, 109, 6904-6909.

46.Chaneton, B.; Hillmann, P.; Zheng, L.; Martin, A.C.; Maddocks, O.D.; Chokkathukalam, A.; Coyle, J.E.; Jankevics, A.; Holding, F.P.; Vousden, K.H. Serine is a natural ligand and allosteric activator of pyruvate kinase M2. Nature 2012, 491, 458-462.

  1. Page 9, first paragraph. Although the αKG generated from serine synthesis may promote stemness, it is important to point out that there are multiple publications showing that αKG can induce differentiation in tumor cells 2-3.

We thank the reviewer for this valuable comment. We now included the two papers suggested by the reviewer, references 105 and 106 in page 11, text line 455-467, pointing out the opposing roles that αKG fulfills in stemness and differentiation. The reference 105 also describes the dependency on exogeneous serine rather than de novo serine/glycine synthesis pathway, and therefore highlight the need for future research in this field to further understand serine/glycine metabolism in (cancer) stem cells and treatment response.

  1. Page 9, second paragraph. There is no evidence showing that one carbon unit from serine is used for SAM synthesis. Ref 85 showed that (PKC)λ/ι KO increased M+1 mCyt labeling from 13C-methinonine, not from serine.

We included a more detailed explanation of this reference, now reference 104 in page 11, text lines 446-454, to avoid the misunderstanding of evidence showing that one-carbon units from serine are used for SAM synthesis.

Reviewer 2 Report

The review article by Sanchez-Castillo and colleagues provides a unique perspective on the serine/glycine pathway’s role in cancer and is of benefit to the scientific community. Although several review articles exist that discuss the signaling and role of ser/gly in cancer in general, this manuscript represents the next step in the application of our understanding-to improve therapy. Several suggestions are written below to improve the quality of this work.

General comments

  • The quality of the English writing could be improved.
  • Given the uniqueness of the focus of this review, it would be better served if the title were not just focused on radiotherapy, but instead created a focus on the role of ser/gly in anti-cancer therapy in general. In so doing, the organization of the paper could be modified to still include the first section focused on general ser/gly signaling and its role in cancer (obligatory to orient the reader) but then create sections based on different types of therapies and how ser/gly has been shown to be or predicted to be, involved in resistance to these therapies-while expanding on each section. For instance, sections on chemotherapy, drugs targeting signaling pathways, oncogenes, etc. Of course a section should be dedicated to radiotherapy as well, as it stands. Certainly, the sub-sections used for the radiotherapy section could be added to sections concerning other therapies. Perhaps these other therapies could be incorporated into Fig. 4. As well.
  • Section 5-in addition to incorporating future perspectives of other therapies (as described in point 2 above), mention of how to practically restrict ser/gly in humans should be addressed, especially in diet. The utility of this approach has of course been commented on in in vitro and in vivo pre-clinical models, but should be addressed for the human, even if speculative.

Specific Comments

  • In several places where the authors describe a concept, too often a review article is used when the original research article(s) should be cited.
  • Some key papers are not referenced but should be. Those are listed below. Further, the last 3 shown below speak to the importance of the contributions of exogenous and endogenous sources of serine, which should be a concept mentioned in the current manuscript-especially in light of combining this strategy with treatment. Also, the paper by Montrose et al., shows relevance for colorectal cancer, which could be mentioned in light of the other cancers discussed, but more importantly work with 5-FU in that paper is directly in line with the main focus of this review and should be mentioned.

Labuschagne, Cell Reports, 2014

Maddocks, Nature 2012

Gao, Cell Reports, 2018

Mendez-Lucas Nature metabolism, 2020

Montrose et al. Cancer Research, 2021

Sullivan, Cell Metab, 2019

  • The right side of Figure 1 (blocks showing different classes of molecules) could be better displayed. Further, writing ‘synthesis sphingolipids’ or ‘synthesis proteins’, etc, doesn’t make sense grammatically.
  • Figure 2 is confusing and needs to be better described in the legend. What is meant by ‘Pathway Alteration’? In the legend, ‘pathway activation’ is written, do the authors mean to write this instead, in the graphs themselves?
  • Figure 5- It is hard to imagine how informative circulating serine/glycine levels in patients would be, given that they may differ dramatically with diet and it would not inform on whether the AAs are actually getting into the tumors. It may be more useful to evaluate the expression of serine synthesis enzymes and/or transporter levels in the tumors themselves. The use of the term ‘mitochondrial inhibitors’ is too vague. Which exact compounds would be expected to be useful in this setting?

Author Response

Reviewer 2

General comments

  1. The quality of the English writing could be improved.

We have checked the manuscript for English spelling and made some changes to make the manuscript more readable.

  1. Given the uniqueness of the focus of this review, it would be better served if the title were not just focused on radiotherapy, but instead created a focus on the role of ser/gly in anti-cancer therapy in general. In so doing, the organization of the paper could be modified to still include the first section focused on general ser/gly signaling and its role in cancer (obligatory to orient the reader) but then create sections based on different types of therapies and how ser/gly has been shown to be or predicted to be, involved in resistance to these therapies-while expanding on each section. For instance, sections on chemotherapy, drugs targeting signaling pathways, oncogenes, etc. Of course a section should be dedicated to radiotherapy as well, as it stands. Certainly, the sub-sections used for the radiotherapy section could be added to sections concerning other therapies. Perhaps these other therapies could be incorporated into Fig. 4. As well.

We thank the reviewer for this suggestion. We agree that the inclusion of other anti-cancer therapies might be of great interest, however, we recently published a Nature Metabolism review where these can be found. This review is written in request to an invite for the special issue in this journal: Advances in Experimental Radiotherapy. The invite and purpose of this review is to focus on the link between serine/glycine and radiotherapy resistance mechanisms. As such, we more clearly refer to other cancer therapies by referencing our latest review on this subject in section 3 of this manuscript in text line 258-355. 

  1. Section 5-in addition to incorporating future perspectives of other therapies (as described in point 2 above), mention of how to practically restrict ser/gly in humans should be addressed, especially in diet. The utility of this approach has of course been commented on in in vitro and in vivo pre-clinical models, but should be addressed for the human, even if speculative.

We thank the reviewer for this suggestion. We include some comments about the applicability of dietary intervention in page 17, text lines 723-728.

Specific Comments

  1. In several places where the authors describe a concept, too often a review article is used when the original research article(s) should be cited.

We agree with this remark and we have changed the manuscript accordingly as much as possible.

  1. Some key papers are not referenced but should be. Those are listed below. Further, the last 3 shown below speak to the importance of the contributions of exogenous and endogenous sources of serine, which should be a concept mentioned in the current manuscript-especially in light of combining this strategy with treatment. Also, the paper by Montrose et al., shows relevance for colorectal cancer, which could be mentioned in light of the other cancers discussed, but more importantly work with 5-FU in that paper is directly in line with the main focus of this review and should be mentioned.

We thank the reviewer for this valuable comment. We have included all the references mentioned by the reviewer: references 15, 25, 46, 77, 156 and 157.

Reference 25 (Gao et. al, 2018) has been explained in page 17, text line 750, to illustrate the functions of serine metabolism in the mitochondria and to support the studies that show synergistic antitumor effects using serine-depleted diets and inhibition of de novo serine/glycine biosynthesis in combination with mitochondrial inhibitors.

References 156 and 157 (Mendez-Lucas Nature metabolism, 2020 and Sullivan, Cell Metabolism, 2019) have been explained in section 5 page 17, text line 705-720, to show the relevance of serine/glycine dietary intervention and inhibition of de novo serine/glycine biosynthesis as combination therapy in cancer patients.

Reference 77 (Montrose et al., Cancer Research, 2021) is included in page 7 as part section 3, text line 320-335, illustrating the relevance of endogenous and exogenous serine in the resistance to 5-FU, as well as in section 5 emphasizing the consideration of serine/glycine dietary interventions and inhibition of de novo serine/glycine biosynthesis to enhance antitumor activity in CRC.

  1. The right side of Figure 1 (blocks showing different classes of molecules) could be better displayed. Further, writing ‘synthesis sphingolipids’ or ‘synthesis proteins’, etc, doesn’t make sense grammatically.

We improved the figure accordingly.

  1. Figure 2 is confusing and needs to be better described in the legend. What is meant by ‘Pathway Alteration’? In the legend, ‘pathway activation’ is written, do the authors mean to write this instead, in the graphs themselves?

The legend of this figure has been modified for best comprehension. The graphs represent the alterations of serine/glycine biosynthesis genes that might be associated with pathway activation in these cancer types, i.e., increased mRNA expression, somatic mutations and copy number gain/amplification.

  1. Figure 5- It is hard to imagine how informative circulating serine/glycine levels in patients would be, given that they may differ dramatically with diet and it would not inform on whether the AAs are actually getting into the tumors. It may be more useful to evaluate the expression of serine synthesis enzymes and/or transporter levels in the tumors themselves. The use of the term ‘mitochondrial inhibitors’ is too vague. Which exact compounds would be expected to be useful in this setting?

We thank the reviewer for these comments. The figure has been modified, including enzyme histology expression levels and we have added this observation about serine levels in section 5, text lines 720-723, in which we mention that serine levels that are available for tumors in the microenvironment depends not only on the plasma metabolite levels but also the capacity of tumors to obtain these nutrients, which is influenced, for instance, by the vasculature or metabolic competition with stromal and immune cells.

Regarding mitochondrial inhibitors, more information has been included in text lines 730-772, speculating about the use of metformin and artemether in combination with radiotherapy and serine/glycine metabolism-targeted therapies to radiosensitize different tumor types.

Reviewer 3 Report

In this review, the authors present the knowledge’s on serine metabolism, cancer proliferation and resistance to therapies.

The review is well written and very interesting in the field. The review can be publish in Cancers without hesitation.

In the first sentence of the introduction "hallmark of carcinogenesis". carcinogenesis should be change by tumorigenesis. Carcinoma are not the only tumors with metabolic modifications.

part 2.2 maybe the first and second paragraph should be invert for a best comprehension.

The authors should complete the figure 1 with the reversible reactions. Maybe it will be pertinent to add the redox response.

Add the end of the p5. The authors didn’t give any reference. They should add in the paper the 2 references below:

  • Serine synthesis pathway inhibition cooperates with dietary serine and glycine limitation for cancer therapy. Tajan et al. nature communication (2021) 12:366
  • Targeting MDM2-dependent serine metabolism as a therapeutic strategy for liposarcoma . Cisse et al. Sci. Transl. Med. 12, eaay2163 (2020)

At the end of the p6, punctuation fails between “pathway and Significantly”

The authors could speak about PHGDH inhibitor and add this inhibitor on the figure 5.

Author Response

Reviewer 3

  1. In the first sentence of the introduction "hallmark of carcinogenesis". carcinogenesis should be change by tumorigenesis. Carcinoma are not the only tumors with metabolic modifications.

The term carcinogenesis has been now changed to tumorigenesis.

  1. part 2.2 maybe the first and second paragraph should be invert for a best comprehension.

We thank the reviewer for this suggestion. The sentence from the second paragraph “The number of cancer types identified as dependent on de novo serine/glycine biosynthesis is growing. Activation of the de novo serine/ glycine biosynthesis pathway has been observed in cancer patients both as a consequence of gene amplification and alterations of upstream regulators that promote enhanced enzyme overexpression” has been now used as introduction for this section in page 3.

  1. The authors should complete the figure 1 with the reversible reactions. Maybe it will be pertinent to add the redox response.

We now included the reversible reactions and redox response.

  1. Add the end of the p5. The authors didn’t give any reference. They should add in the paper the 2 references below:
  • Serine synthesis pathway inhibition cooperates with dietary serine and glycine limitation for cancer therapy. Tajan et al. nature communication (2021) 12:366
  • Targeting MDM2-dependent serine metabolism as a therapeutic strategy for liposarcoma. Cisse et al. Sci. Transl. Med. 12, eaay2163 (2020)

We thank the reviewer for this comment. The paper (Tajan et al., 2021) was already included in section 5, text line 597, now reference 155, in which we wanted to illustrate the additive antitumor function of serine/glycine biosynthesis pathway inhibition and dietary restriction of serine and glycine. We included the paper (Cisse et al, 2020) in page 7, text lines 244-252, reference 55, to introduce MDM2-mediated regulation of serine metabolism in liposarcomas and cited this paper again in section 5, page 17, text line 712, to also show that serine/glycine-deprived conditions additively enhanced the antitumor effect of MDM2 and PHGDH inhibition.

  1. At the end of the p6, punctuation fails between “pathway and Significantly”.

The typo has been corrected.

  1. The authors could speak about PHGDH inhibitor and add this inhibitor on the figure 5.

We thank the reviewer for this suggestion. We included PHGDH inhibitors in the figure 5. However, in this review we did not explain in detail the different inhibitors of serine/glycine synthesis pathway.  PHGDH and SHMT1/2 inhibitors and limitations have been recently reviewed in Geeraerts, S.L et al, 2021, reference 11.

  1. Geeraerts, S.L.; Heylen, E.; De Keersmaecker, K.; Kampen, K.R. The ins and outs of serine and glycine metabolism in cancer. Nature Metabolism 2021, 10.1038/s42255-020-00329-9, doi:10.1038/s42255-020-00329-9.

Round 2

Reviewer 2 Report

The authors have addressed my concerns.